# The Zooxanthellate Jellyfish Holobiont *Cassiopea andromeda,* a Source of Soluble Bioactive Compounds

**DOI:** 10.3390/md21050272

**Published:** 2023-04-26

**Authors:** Stefania De Domenico, Gianluca De Rinaldis, Marta Mammone, Mar Bosch-Belmar, Stefano Piraino, Antonella Leone

**Affiliations:** 1Istituto di Scienze delle Produzioni Alimentari, Consiglio Nazionale delle Ricerche (CNR-ISPA, Lecce), 73100 Lecce, Italy; 2Dipartimento di Scienze e Tecnologie Ambientali, Università del Salento, 73100 Lecce, Italy; 3Istituto di Nanotecnologia, Consiglio Nazionale delle Ricerche (CNR-NANOTEC), 73100 Lecce, Italy; 4Dipartimento Scienze della Terra e del Mare, Università degli Studi di Palermo, 90133 Palermo, Italy; 5Research Unit Lecce, Consorzio Nazionale Interuniversitario per le Scienze del Mare (CoNISMa), 73100 Lecce, Italy; 6National Biodiversity Future Center (NBFC), S.c.a.r.l., 90133 Palermo, Italy

**Keywords:** jellyfish extracts, Symbiodiniaceae, non-indigenous species, antioxidant activity, pigments, bioactive compounds

## Abstract

*Cassiopea andromeda* (Forsskål, 1775), commonly found across the Indo-Pacific Ocean, the Red Sea, and now also in the warmest areas of the Mediterranean Sea, is a scyphozoan jellyfish that hosts autotrophic dinoflagellate symbionts (family Symbiodiniaceae). Besides supplying photosynthates to their host, these microalgae are known to produce bioactive compounds as long-chain unsaturated fatty acids, polyphenols, and pigments, including carotenoids, with antioxidant properties and other beneficial biological activities. By the present study, a fractionation method was applied on the hydroalcoholic extract from two main body parts (oral arms and umbrella) of the jellyfish holobiont to obtain an improved biochemical characterization of the obtained fractions from the two body parts. The composition of each fraction (i.e., proteins, phenols, fatty acids, and pigments) as well as the associated antioxidant activity were analyzed. The oral arms proved richer in zooxanthellae and pigments than the umbrella. The applied fractionation method was effective in separating pigments and fatty acids into a lipophilic fraction from proteins and pigment–protein complexes. Therefore, the *C. andromeda*–dinoflagellate holobiont might be considered as a promising natural source of multiple bioactive compounds produced through mixotrophic metabolism, which are of interest for a wide range of biotechnological applications.

## 1. Introduction

The phylum Cnidaria is a diverse and ecologically important group of marine invertebrates. Among them, several members of the class Scyphozoa receive increasing attention due to the conspicuous size of their adult medusa stage (usually known as jellyfish) and their recurring population outbreaks, which may negatively affect human activities at sea, including leisure and fisheries [1]. Besides the biological and ecological mechanisms boosting their proliferations, several jellyfish species are investigated worldwide as sources of bioactive compounds [2,3,4,5,6,7]. Further, some Mediterranean jellyfish have been considered as novel foods in Western countries [2,8,9,10,11,12].

Cnidarians are often associated with microbes [13], including tight functional symbiotic relationships collectively termed as holobionts. Cnidarian holobionts include bacteria, archaea, fungi, and viruses [14] and some unicellular dinoflagellates of the family Symbiodiniaceae that generate photosymbiosis (an association between heterotrophic hosts and microalgae) [15,16]. The holobiont is an active part in biogeochemical processes, and there is a close relationship between host and microorganism for nutrient assimilation and cycling. The association allows the holobiont to directly access the autotrophically fixed carbon and therefore to obtain advantages in oligotrophic environments with limited prey availability. In some scyphozoans, the photosynthetic apparatus of the endosymbiotic algae can support the energy needs of the jellyfish host [17], which in turn provides protection to the microalgae and optimal conditions with respect to light, CO_2_, and nutrient supply. Although the symbioses between zooxanthellae and medusozoa are often poorly documented in the literature, 20–25% of Scyphozoa species are zooxanthellate (including facultative symbiotic species), and the suborder Kolpophorae (Scyphozoa: Rhizostomeae) consists, with few exceptions, of only zooxanthellate jellyfish. Of particular interest are organisms adapted to high-nutrient environments, such as the obligate photosymbiotic jellyfish *Cassiopea* spp., which are commonly found in mangrove habitats, seagrass meadows, reefs, sheltered lagoons, and harbors.

*Cassiopea andromeda* (Forsskål, 1775) (Cnidaria: Scyphozoa: Rhizostomeae) is a jellyfish native to the Indo-Pacific region and a well-known Lessepsian immigrant that, more than 120 years ago, entered the Mediterranean Sea through the Suez Canal and slowly spread westwards and is recurrently recorded in some harbors of Sicily [18,19]. This species has a high invasive potential [20] and, similarly to other outbreak-forming jellyfish, might have significant ecological impacts on the Mediterranean food webs [21]. Indeed, *C. andromeda* has an advantageous mixotrophic metabolism [22] due to the occurrence of endosymbiotic dinoflagellates of the family Symbiodiniaceae; because of this, the jellyfish was found to contain compounds (in some cases easily extractable) with valuable biological activities [7].

The analysis of the biochemical composition of most Scyphomedusae indicates that 95–97% of the fresh weight is made by water; however, proteins (mainly collagen) are the main organic macromolecules, while non-proteinaceous compounds such as carbohydrates and lipids represent minor components. Moreover, compounds such as phenols and pigments, mainly derived from dinoflagellates, are also detected in zooxanthellate jellyfish [3,4,7,23,24]. Members of the order Rhizostomeae have a higher protein content than Semeostomeae; however, the percentage of minority compounds can be very different among jellyfish species also depending on the co-presence of microorganisms [4,6].

A growing interest in compounds with biological activity has indicated that both majority and minority components of jellyfish can deserve consideration as active compounds useful in biotechnological applications. Partially purified compounds and extracts have antiproliferative properties against human cancer cells [4,25]. Several jellyfish proteins and peptides have been also studied, showing antioxidant capacity and other important biological effects [3,4,5,6,26,27,28,29]. In general, studies were focused on the proteinaceous compounds of jellyfish, such as collagen and collagen hydrolysate (collagen peptides), that have been shown to exert several effects such as immunomodulatory, antioxidant, and wound healing ability [5,30,31]. The zooxanthellate species *Cotylorhiza tuberculata* (Macri, 1778), one of the most common jellyfish in the Mediterranean Sea, hosts endosymbiotic zooxanthellae (Symbiodiniaceae) producing polyphenols, proteins, and other antioxidant compounds. The extracts of this jellyfish exert antioxidant ability and specific cytotoxicity towards MCF-7 breast cancer cells but not towards non-cancer HEKa cells [4].

Among the bioactive compounds, greater emphasis has been placed on natural antioxidants that appear to be effective in the prevention of many chronic diseases including cancer, whose pathogenetic basis is often linked to imbalance of reactive oxygen and nitrogen species (ROS/RNS). Phytochemicals with antioxidant activity such as carotenoids and polyphenols might prevent cancer and other human diseases [32,33]. Carotenoids are a diverse group of pigments that are essential in photosynthetic organs and organisms along with chlorophylls. The unique structure of these compounds determines their potential biological functions and actions. The pattern of conjugated double bonds in the polyene backbone of carotenoids determines their light-absorbing properties and influences their antioxidant ability [34]. Carotenoids also act as photo-protectants, antioxidants, color attractants, and precursors of plant hormones in non-photosynthetic organs [35]. Carotenoids and their by-products perform various biological functions in the growth, development, and reproduction of animals and are essential nutrients for humans. However, with the exception of a few arthropods [36], metazoans are not able to synthesize carotenoids de novo but can obtain them directly from food and can partially modify them through metabolic reactions. The well-known antioxidant activity of phenolic compounds largely depends on the amount of their phenol rings (aromatic ring) and the hydroxyl groups (functional groups) [37]. Their activity is again related to their ability to scavenge reactive oxygen species (ROS), which are naturally produced in organisms. Polyphenols are now well recognized as protectors against cancer and other diseases [38,39,40]. Moreover, polyphenols are able to act on cell signaling [41,42], growth inhibition [43,44], and apoptosis induction [45,46,47] and as modulators of enzymatic activity [48]. However, there are still few data on jellyfish polyphenols in the scientific literature [3,5,49].

Overall, the biotechnological potential of jellyfish has been relatively underexploited, so far with a limited number of jellyfish-derived products.

Based on the hypothesis that *C. andromeda* may represent a promising organism for bioprospecting and also considering its ease of growth and breeding [50], we previously set up protocols for protein isolation and for the extraction of soluble compounds [7]. We analyzed the whole dried biomass of *C. andromeda*, demonstrating that the hydroalcoholic extracts exerted a remarkable antioxidant activity mainly in the oral arms [7]. To deepen the study and characterization of *C. andromeda* and its possible applications, here, we focused on the hydroalcoholic soluble compounds by further sub-fractionation and characterization of the hydroalcoholic extracts separately obtained from the umbrella and oral arms of *C. andromeda.* In particular, we investigated (a) the specific composition in proteins, lipids, pigments, and phenolic compounds of each fraction and body part; (b) the in vitro antioxidant activity of each the fractionated hydroalcoholic extract; and (c) the relative concentration of dinoflagellate symbionts and of the pigment contents in the two jellyfish body parts. Overall, our findings corroborate the hypothesis that the *C. andromeda* holobiont has a high potential to become a source of molecules with various biological activities and biotechnological applications.

## 2. Results and Discussion

A gross biochemical characterization of *C. andromeda* (whole jellyfish and separate body parts) was previously carried out by the development and implementation of a specific extraction protocol [4,7]. The main goal of this work was to further characterize—by an ad hoc fractionation protocol [4]—the hydroalcoholic extract, whose fractions concurrently demonstrated strong biological activity (Leone, manuscript in preparation).

### 2.1. Hydroalcoholic (80% Ethanol) Extraction

To investigate the biochemical composition and the biological activity of the different jellyfish tissues, the extractions of soluble compounds by hydroalcoholic solution were performed from freeze-dried samples of the umbrella (UMB) and oral arms (OA) of eight specimens of *C. andromeda*, sampled in the harbor of Palermo (NW Sicily). Hydroalcoholic-soluble compounds were extracted with an 80% ethanol solution, as successfully adopted in previous works [4,7]. The total freeze-dried extracts (ExDW) obtained, expressed as grams of dry weight and as percentage of the lyophilized whole jellyfish samples, are showed in Table 1.

No difference among the different jellyfish specimens was found, as a quite constant amount of hydroalcoholic extract was obtained from the different samples. The amount of dried extract from UMB samples was in mean 0.300 ± 0.08 g, which was about 42.1% of the DW of the starting biomass, while the amount of extract from OA was higher than the UMB extracts with a mean of 0.449 ± 0.116 g, which was about the 44.7% of the dried initial biomass (Table 1).

The gross biochemical characterization of the *C. andromeda* biomass [7] previously revealed a remarkable amount of antioxidant compounds mainly from digestible proteins, lipids, and phenolic compounds and a higher concentration of polyunsaturated fatty acids and pigments than in non-zooxanthellate jellyfish. Further, De Rinaldis et al. (2021) evidenced a high antioxidant activity in the hydroalcoholic extract, mainly from the oral arms. Here, we set up a protocol to further separate the complex hydroalcoholic extract and to characterize its bioactive components.

### 2.2. Fractionation of the Hydroalcoholic Extracts and Biochemical Characterization of Fractions

The total jellyfish hydroalcoholic extracts (80% ethanol) contain a mixture of heterogeneous compounds. In order to obtain partially purified fractions, a first separation was performed by protein precipitation. A 50% acetonitrile (ACN) solution was used to efficiently precipitate proteins larger than ~40 kDa [4]. This phase-separation methodology was used for the first time to partially characterize the complex hydroalcoholic extract of the whole jellyfish *Cotylorhiza tuberculata* [4]. This method is useful for separating different classes of components eventually present in hydroalcoholic-soluble extracts of complex samples, such as biomass of zooxanthellate jellyfish. Indeed, the use of a mild organic solvent (FDA, 2012) such as the solution acetonitrile/water (ACN:H_2_O) and the avoidance of freeze–thaw cycles, which are damaging the structures of bioactive compounds, resulted in an advantageous method for composite organisms such as holobionts [4].

After resuspension of ExDW in acetonitrile/water solution, a three-phase separation was obtained, similar to that observed in *C. tuberculata* samples [4]. A lipophilic upper phase (UP) and a hydrophilic lower phase (LP) were separated by a semisolid, green-colored, intermediate phase (IP) (see Materials and Methods, Figure 6).

After an accurate separation, the characterization of the three phases of the jellyfish extract was carried out.

Semi-solid IP fractions were bright green, probably due to pigment–protein complexes of symbiotic zooxanthellae, as evidenced for IP and LP in the jellyfish *Cotylorhiza tuberculata* [4]. The unique consistency of the intermediate phase (IP) was not useful for accurate analysis due to the insolubility in most of the solvents suitable for the study of biological activity on cell culture systems; therefore, the IP fractions of both the umbrella and the oral arms were not considered for biochemical characterization in this work.

The lipophilic upper phase (UP) and the aqueous lower phase (LP) were characterized for protein and phenol contents and for their antioxidant activity.

#### 2.2.1. Spectroscopic Analysis: Absorbance and Fluorescence Spectra

Given the lack of literature data on the possible compounds present in the hydroalcoholic extract fractions and of related analyses for their identification, a simple qualitative analysis of the upper phase (UP) and the lower phase (LP) was first performed to rapidly identify compound categories rather than individual compounds. Both the absorbance and fluorescence spectra were measured (from ultraviolet to near infra-red region) for both the upper and lower phases from OA and UMB samples of jellyfish. The absorbance spectra of both the upper phases (umbrella and oral arms) were similar (Figure 1) and revealed three different absorbance regions: a high-absorbance region from 230 to 280 nm where peptide bonds in proteins and free amino acids have maximum absorption [51,52], a broad-shaped peak at 453 nm (400–500 nm), and a third, weaker absorption peak at 663 nm (650–675 nm). The last two peaks correspond to the characteristic absorption spectra of the photosynthetic pigments and are related to chlorophylls, carotenoids, and pigment–protein complexes [53,54]. The absorbance spectra of the lower phases of UMB and OA were also similar to each other, revealing two peaks at 280 nm and 329 nm (310–340 nm). Although the compounds were not identified by this analysis, these spectra fall within the UV absorbance ranges of subfamilies of phenolics, i.e., phenolic acids (270–280 nm and 305–330 nm), and flavonoids (270–280 and 310–350 nm).

Fluorescence spectra (Figure 2) revealed a difference between the upper and lower phases of both the extracts from the *C. andromeda* umbrella and oral arms. Both the upper phases (UP-UMB and UP-OA) have a fluorescence emission between 280 and 380 nm, which was much less evident for LPs. Fluorescent values were of 1250 and 1000 RFU (relative fluorescence units) for the upper phases of the umbrella and oral arms, respectively, and about 125 and 150 RFU for the lower phases of the umbrella and oral arms, respectively. Wolf and Stevens (1967) [55] reported that carotenoid such as β-carotene and lutein emitted fluorescence in the range 300–400 nm upon excitation at 280 nm; however, wide varieties of natural molecules are characterized by fluorescence in these regions of the electromagnetic spectrum [56]. In addition, some synthetic compounds such as polycyclic aromatic hydrocarbons (PAH) or pesticides emit fluorescence in the same range under excitation at a wavelength of 280 nm, and they can be easily adsorbed by the benthic *Cassiopea andromeda*, especially from polluted sediments such as those of port seabeds [57,58,59]. Analyses for the identification of the involved chemical species are still in progress in order to identify both useful and dangerous compounds. Weak peaks (less than 25–50 RFU) were also detectable in the range 650–700 nm, which is typical of a fluorescence spectrum of chlorophyll-a [56,60].

#### 2.2.2. Protein Content

In a previous work [7], we quantified soluble and insoluble proteins from whole tissues of the umbrella and oral arms of the holobiont *C. andromeda*. Here, we focus on the fractions of the hydroalcoholic extract, as it demonstrates an interesting biological activity. Table 2 shows the concentration of proteins detected in the UP and LP phases from both the umbrella (UMB) and oral arms (OA), expressed as mg of proteins per gram of dried hydroalcoholic extract (ExDW). Protein content in the UP (lipophilic fraction) of *C. andromeda* umbrella was estimated to be on average 1.71 ± 0.94 mg protein per gram of extract dried weight, while in the LP (hydrophilic fraction), it was 4.95 ± 2.87 mg/g of ExDW. The mean protein content of UP and LP from the oral arm extract was 4.33 ± 1.60 and 6.71 ± 3.58 mg of protein per gram of ExDW, respectively (Table 2). A large data variability was observed, probably due to the samples coming from two different sampling sites and being sampled at two different times. Despite the variability of the samples, the data clearly indicated a significant difference between the protein content of UP and LP, showing that when only UP and LP are considered (ignoring the intermediate fraction), the protein compounds are mostly separated in the lower fraction. Data from each specimen showed that the fractions obtained from the extracts coming from the oral arms’ samples gave a higher protein content as compared to the fractions coming from the umbrellas’ extract. However, the data taken all together did not indicate significant differences between the protein content of UPs and LPs from UMB and OA.

#### 2.2.3. Phenolic Compounds

The hydroalcoholic solvent allowed the extraction of a large number of phenolic compounds from *C. andromeda* [7]. To characterize the *C. andromeda* hydroalcoholic extract derived from different jellyfish body parts, the content of total phenols was measured in the two fractions, namely UP and LP, derived from both the UMB and OA of six different specimens (Table 3). Similar to the protein content, some variability was also found in the phenolic content. However, the concentration of phenol compounds was significantly higher in LPs as compared to UPs in all specimens, indicating a specific separation of compounds by this protocol, as also found in *Cotylorhiza tuberculata* samples [4] and as can be assumed from the absorbance spectra of the LPs (Figure 1). Indeed, in the umbrella tissues, the phenolic compounds were 3–4 times higher in LP than in UP regardless of the amount extracted in the different specimens. The concentration of phenol compounds was significantly higher in the hydroalcoholic extract fractions (UP and LP) of the oral arms as compared to the umbrella body part, confirming that the hydroalcoholic soluble compounds present in the oral arms are at least quantitatively different from that in the umbrella [7]. Phenolic compounds are widespread in plants, algae, and microorganisms and are one of the most effective antioxidants known in nature [61]. The presence of phenol compounds in *C. andromeda* is likely related to the presence of endosymbiotic zooxanthellae. The detected variability in the content of total phenols could be also related to the different zooxanthellae species or other undetected microorganisms associated with jellyfish in nature other than the inherent genetic variability of specimens. Moreover, as microalgae are the primary producers of phenols [62], it could be suggested that the greater quantity of phenols found in the extract fractions of the oral arms compared to the umbrella might be attributed to either a varying number of zooxanthellae or to the photosynthetic activity of the endosymbiotic zooxanthellae. This might be due to the fact that this particular jellyfish has a tendency to remain upside down on the sea floor, which results in better exposure to light for the photosynthetic algae situated in the oral arms [63]. Phenolic compounds from exogenous origin should be also considered and are currently under evaluation.

### 2.3. Antioxidant Activity

The antioxidant activity (AA) was evaluated in the fractions UP and LP of the hydroalcoholic extracts (Table 4). In order to verify the effectiveness of the fractionation protocol in segregating compounds belonging to different chemical species and, presumably, with different antioxidant activity, the AA values were expressed as nanomoles of Trolox equivalents per g of jellyfish tissues (dry weight of UMB or OA) and per dried hydroalcoholic extract (Table 4). Despite the high inter-sample variability, as indicated by the high standard deviation, clear trends were identified. Notably, the results in Table 4 showed significantly higher AA values in the LP than in the UP fractions in both UMB and OA.

Since the LP showed the higher antioxidant capacity as compared to UP and, similarly, both proteins and phenols are more concentrated in LP, we can infer that AA is related to protein compounds, amino acids, as well as active phenolic groups and the high reactivity of these species.

No significant difference between the antioxidant activity of UMB and OA was found, suggesting that the tissues of this holobiont are quite homogeneous in terms of antioxidant compounds extractable by hydroalcoholic solution. To verify whether the antioxidant compounds come mainly from endosymbiotic microalgae, as might be hypothesized, the density of zooxanthellae in UMB and OA was analyzed.

### 2.4. Density of Zooxanthellae in Umbrella and Oral Arms

To determine whether there are any dissimilarities in the density of zooxanthellae or their pigment content in the tissues of the umbrella or oral arms, automated systems were employed to count the autofluorescent zooxanthellae. The results refer to the dry weight (DW) of the jellyfish tissue, as described in the Materials and Methods section.

In Appendix A, representative pictures of the density of microalgae symbiont in the umbrella (A) and oral arm tissues (B) are shown. Figure 3 shows the number of red-fluorescent cells, green-fluorescent particles, and cells showing both red and green autofluorescence (Figure 3A). Figure 3B is a representative picture of the lyophilized tissue resuspended in seawater and detected by confocal microscopy. Microalgae (m) containing the red-autofluorescent chloroplasts are evident as well as a number of heterogenous particles (p), including some cnidocytes, also showing green autofluorescence; an untriggered cnidocyst is also visible (c). In addition to the red-autofluorescent chloroplast, several microalgae show a corpuscle with a bright-green autofluorescence inside the cell (arrow in Figure 3B), as also found in *C. tuberculata* [4]. Only the round-shaped red-autofluorescent cells with a diameter around 10 μm were considered as microalgal cells and considered in the counting. The number of zooxanthellae was significantly (*p* < 0.001) higher in the oral arms (OA) than in the umbrella (UMB) tissues. This is obviously due to the phenotypic trait of the *C. andromeda*, which exposes the oral arms to the light and also may justify the different content in proteins and phenol compounds detected in the oral arms as compared to umbrella tissues (Table 2 and Table 3). However, in our experimental conditions, the difference in zooxanthellae amount between UMB and OA is not supported by a significant difference in antioxidant activity (Table 4).

### 2.5. Fatty Acid Profile in Lipophilic Fraction (Upper Phase) of the 80% Ethanol Extract

In a previous work [7], a total lipid extraction was carried out from the whole dried biomass of *C. andromeda* and from its hydroalcoholic extract (80% ethanol) by chloroform:methanol extraction [7]. We demonstrated that *C. andromeda* contained an appreciable amount of total lipids (about 1% of the DW of whole freeze-dried jellyfish) and that a total lipid amount of 0.62% DW of whole jellyfish could be obtained by extraction with 80% ethanol (corresponding to 15.5 ± 0.5 mg per gram of hydroalcoholic extract) [7].

Here, the fractionation of the 80% ethanol extract allowed isolation of the lipophilic fraction containing semi-polar and non-polar compounds in the UP. This study revealed that the use of acetonitrile/water fractionation on the hydroalcoholic extract was successful in isolating a total of 10.2 ± 0.7 mg of lipids per gram of hydroalcoholic extract in the UP fraction. This indicates that this fractionation technique is a reliable enrichment method.

In order to characterize the fatty acid (FA) profiles in the UP fraction, a GC-MS analysis was carried out. The fatty acids (FAs) composition of the lipophilic upper phase (UP) of the hydroalcoholic extract of *C. andromeda* biomass is shown in Table 5.

Most FAs belong to the class of polyunsaturated FAs (PUFA, 54.4 ± 5.5%) and saturated FAs (SFA, 39.8 ± 0.4%), while only 5.8 ± 0.8% were monounsaturated FAs (MUFA). The data show a similar FA composition in the UP compared with the original hydroalcoholic extract [7], with a little enrichment in SFAs as compared to the starting hydroalcoholic extract, where SFA were about 31% [7], and a slight loss of MUFA (63%, respectively, in De Rinaldis et al., 2021). This may be due to the greater stability of SFAs during the fractionation and purification process as compared to double-bonds-containing FAs.

Among the SFAs, the palmitic acid (C16:0) was the most representative (about 25%), confirmed to be the dominant SFA in zooxanthellate *Cassiopea* jellyfish, as reported by [63]. In our conditions, lauric (C12:0, 8.5%), stearic (C18:0, 3.6%), and myristic (C14:0, 2.5%) acids were also detected (Table 5) as the whole biomass of the holobiont was extracted.

Isoleic acid (C18:1 trans-10), a little-known, long trans-fatty acid, and palmitoleic acid (C16:1) were the only MUFAs detected in the UP, with enrichment in isoleic acid in the UP as compared to the EtOH extract of *C. andromeda* [7].

The more representative PUFAs in UP were stearidonic (C18:4ω3, 16.3%), arachidonic (C20:4ω6, 14.4%), and docosahexaenoic (C22:6ω3, 14.0%) acids, followed by linolenic (C18:3ω3, 2.4%), eicosadienoic (C20:2ω6, 2.7%), eicosapentaenoic (C20:5ω3, 2.4%), and docosapentaenoic (C22:5ω3, 2.2%) acids. All the identified PUFAs are typical FAs of the zooxanthellae, especially C18:4ω3, C20:5ω3, and C22:6ω3 [64]; meanwhile, arachidonic acid (C20:4, ω6) was particularly detected in jellyfish tissue [65]. In particular, stearidonic acid C18:4 (ω3) commonly occurs in marine plants and is relatively abundant in dinoflagellates [66,67,68].

As concerning the biological activity of FAs, only few studies reported a biological activity for the MUFA isoleic acid as a positive correlation between the concentration of trans-10 C18:1 in platelets and the degree of coronary artery disease in humans [69] or the effects on plasma lipids and lipoprotein metabolism in rabbits, with an increase in VLDL-cholesterol and non-HDL/HDL ratio [70]. The biological activity of PUFAs has been made known by different studies. The growth and viability of hormone-independent cancer cells lines are strongly inhibited by most PUFAs, as reported by Bratton et al. (2019) [71] in a human prostate cancer cell line treated with EPA and DHA (C20:5ω3 and C22:6ω3) and AA (C20:4ω6) as well as with LA (C18:2ω6) and ALA (C18:3ω3). Moreover, the UP is enriched in ω3 PUFAs, as shown in Table 5, and recent studies suggest that ω3-PUFAs can be cancer chemopreventive, chemosuppressive, and auxiliary agents for breast cancer therapy [72].

Although in the present work we considered the whole biomass of *C. andromeda*, the extraction and fractionation method presented here allows to enrich the fractions of all those FAs of algal origin or also translocated in the host mesoglea. The richness of the UP fraction in essential PUFAs, whether they are ω3 or ω6, establish a technical improvement as compared to previous results [7], as it is a further way of concentrating the bioactive compounds obtained from the zooxanthellate *C. andromeda* by using mild systems. Indeed, with the current trend towards designing green and sustainable extraction methods of natural products, the use of this protocol based on ethanol and acetonitrile, which is a recommended polar solvent scoring a color coding between yellow and green [73], is feasible.

### 2.6. Pigments Quantification in the Fractions

Based on the results obtained from the spectrophotometric analysis and the related literature data, the total hydroalcoholic extracts (ExDW) and the two derived fractions (upper and the lower phases) of *C. andromeda* samples were also analyzed for the content of the five major pigments characteristic of the microalgae endosymbiont *Symbiodinium*, namely chlorophyll-a (Chl-a), β-carotene, diadinoxanthin, peridinin, and lutein. The HPLC analysis provided relative quantification of the pigments present in the ExDW and in its lipophilic fraction (UP) from UMBs and OAs. In Table 6, the concentrations of the analyzed pigments are shown. In particular, in ExDW, the most concentrated pigment is chlorophyll-a, followed by lutein, peridinin, diadinoxanthin, and β-carotene, which was detectable at low concentrations (about 8 mg/g of ExDW). The hydroalcoholic extract (ExDW) from OA is significantly richer in Chl-a (1960.6 ± 601.3 mg/g of ExDW) and lutein (3066.4 ± 425.0 mg/g of ExDW) than the ExDW from UMB (722.3 ± 205.0 and 441.5 ± 15.1 mg/g of ExDW), while there is an equal distribution of the other pigments diadinoxanthin, peridinin, and β-carotene in both the body parts of jellyfish. In a pigment composition analysis by HPLC, Matsuoka et al. (2012) [74] showed that in the *Symbiodinium* strain Y106, the main pigment was chlorophyll-a, followed by peridinin, which was the most abundant carotenoid, while diadinoxanthin, Chl-c2, and pheophytin-a were minor [74]. After the fractionation by acetonitrile/water and HPLC analysis of UP and LP, the LP fraction from both UMB and OA showed no detectable pigments under our analysis conditions, which was also confirmed by the absorption spectrum (Figure 1), where no peak was detected in the characteristic regions of carotenoids and chlorophylls. While the UP fractions showed an enrichment of all the pigments (Table 6), β-carotene was undetectable in the OMB-UP at our conditions.

To analyze the distribution of these pigments in *C. andromeda* holobiont, a quantification of pigments by number of microalgae was performed. Figure 4 shows the concentration of the pigments expressed per gram (DW) of umbrella or oral arm tissues (Figure 4A) and per number of microalgae (Figure 4B). The data indicated no significant differences in the concentration of each pigment between UMB and OA except for lutein. The xanthophyl lutein appears to be the carotenoid also most represented in hydroalcoholic extracts of jellyfish oral arm tissues compared to UMB tissues (Table 6). The pigment distribution in relation to the microalgae count (Figure 4B) exhibited a comparable pattern, revealing a significantly higher concentration of lutein in the microalgae found in the oral arms compared to those present in the umbrella. Lutein is an antenna pigment located in the light-harvesting complex (LHC) of the photosynthetic mechanism of microalgae and is known to prevent cellular oxidative damage under conditions of high light intensity [75]. Recently, the biosynthesis of lutein in microalgae and its potential modulation have been studied [76].

As other cnidarians, the upside-down jellyfish *Cassiopea* sp. lives in an obligate symbiosis with dinoflagellates of the family Symbiodiniaceae (e.g., *Symbiodinium* spp.) and with other microorganism (bacteria, fungi, and other microalgae). Several studies have focused on the bioactivity of pigments extracted from plants, macroalgae, or microalgae such as the dinoflagellate *Symbiodinium,* whose pigments have been recently deeply analyzed because of their importance in coral bleaching. The great diversity of visible light-harvesting complexes (chlorophylls, carotenes, xanthophylls, phycobiliproteins, etc.) includes a number of bioactive compounds [77]. The pigment composition of *Symbiodinium* is distinct from every other alga in containing the xanthophylls peridinin and its isomers, the chlorophyll a (Chl-a) and chlorophyll c2 (Chl-c2), the pigments β-carotene and the minor xanthophyll dinoxanthin, and its products diadinoxanthin and diadinochrome, as reported by HPLC studies of *Symbiodinium* [74,78,79,80,81]. As regards the carotenoids, other studies about the biological activity of these algal pigments have described anti-tumor and anti-inflammatory activities. The apocarotenoid peridinin isolated from the gorgonian *Isis hippuris* inhibits proliferation and survival of HTLV-1-infected T-cell lines both in vitro and in vivo, showing an anti-ATL effect [82]. Due to their chemical structure that is rich in double bonds and provides them with antioxidant properties, xanthophyll as lutein has shown light-filtering, eye protection [83], cancer-risk-lowering, and anti-inflammatory benefits that could be employed in potential therapeutic uses against several chronic diseases [83,84,85,86]. Studies have indicated that chlorophyll-a possesses various biological activities, including chemo-preventive, antioxidant, and antimutagenic activities [87,88,89]. Furthermore, studies have indicated that chlorophylls in general have been found to possess various biological activities such as the ability to trap mutagens, modulate xenobiotic metabolism, induce apoptosis, and exhibit antimicrobial and anti-inflammatory properties [90].

Therefore, the hydroalcoholic extraction and the subsequent fractionation applied on the *C. andromeda* holobiont is a good compromise for the extraction and enrichment of chlorophylls and carotenoids and possibly other bioactive compounds. Furthermore, avoiding the use of harmful organic solvents is in line with the idea of developing green-solvent-based extraction protocols for safe biomass exploitation.

The characterization carried out on the ethanolic extracts and its derived fractions allows us to confirm that the holobiont *C. andromeda*/dinoflagellate can be regarded as a promising source of bioactive molecules such as proteins, pigments, phenolic compounds, and fatty acids that are easily separable and concentrated in the enriched fractions of hydroalcoholic extract. This protocol, based on relatively green solvents such as ethanol and acetonitrile, could represent a basis for the isolation of functional compounds once their bioactivity has been assayed.

Tests of the biological activity of the fractions isolated and characterized here are in progress for potential nutraceutical, cosmeceutical, pharmaceutical, and/or other biotechnological applications.

## 3. Materials and Methods

### 3.1. Chemicals, Materials, and Instruments

Ethanol, acetonitrile, bovine serum albumin (BSA), ABTS [2,20-azinobis (3-ethylben-zothiazoline-6-sulfonic acid) diammonium salt], 3,4,5-trihydroxybenzoic acid (gallic acid), potassium peroxydisulfate, (±)-6-hydroxy-2,5,7,8-tetramethylchromane-2-carboxylic acid (TROLOX), and Folin and Ciocalteu’s phenol reagent were all purchased from Sigma-Aldrich (Merck Life Science srl, Milan, Italy).

Countess™ 3 Automated Cell Counter (Invitrogen, Carlsbad, CA, USA) and Cell Counting Chamber Slides (Invitrogen, Carlsbad, CA, USA) were purchased from Thermo Fisher Scientific, Waltham, MA, USA. Ninety-six-Well Clear Polystyrene Microplates, round-bottom, were from Corning^®^ (Corning, NY, USA). Bio-Rad Protein Assay Dye Reagent concentrates were purchased from Bio-Rad Laboratories (Munich, Germany). Infinite M200, quad4 monochromator™ detection system was from Tecan group (Männedorf, Switzerland). An Agilent 1100 HPLC instrument was used for the HPLC analyses, and diadinoxanthin, peridinin, and chlorophyll-a standards were purchased from ChromeDex (Los Angeles, CA), while b-carotene and lutein were from Sigma-Aldrich (St. Louis, Missouri, US). Countess 3 Automated Cell Counters and Cell Counting Chamber Slides (Thermo Fisher Scientific Inc., Waltham, Massachusetts, US) were used for microalgae automated counting. A LSM 5 Pascal laser scanning confocal microscope (Carl Zeiss, Munchen, Germany) was used for confocal images of microalgae.

### 3.2. Jellyfish Samples

*Cassiopea andromeda* (Forsskål, 1775) jellyfish were sampled within the harbor “la Cala” of Palermo (Sicily, Italy) in November and December 2017 at the same sites investigated by Maggio et al., who assessed the specific identity of this jellyfish population by COI barcoding [18]. Five specimens were sampled in front of “Calamida” and three in front of “Canottieri” sites, at a depth between 0.5 and 2 m. After biometric measurement (weight and umbrella diameter), the umbrella and oral arms of each specimen were separated, immediately frozen in liquid nitrogen, and stored at −80 °C until lyophilization (Appendix A). Lyophilized samples were stored in tubes at −20 °C until use.

### 3.3. Hydroalcoholic Extraction

Lyophilized tissues samples (umbrella and oral arms) were finely crumbled in a mortar using liquid nitrogen, and the resulting dry powder was subjected to hydroalcoholic extraction [4,7] by stirring in 16 volumes (*w*/*v*) of 80% ethanol by a rotary tube mixer with speed at 25 rpm for 16 h at 4 °C. Samples were then centrifuged at 9000 × g for 30 min at 4 °C, the supernatants were separated from the insoluble material, and an aliquot of each was used for biochemical assays. The supernatant was then concentrated by vacuum rotary evaporator (Buchi R-205 with Vacuum pump V-710 and Vacuum controller V-850) at low temperature, treated under a stream of nitrogen gas to completely evaporate the ethanol, and then lyophilized at –40 °C in a chamber pressure of 0.110 mbar (Freezone 4.5 L Dry System, Labconco Co. Thermo Scientific, Kansas City, MO, USA). All the operations were performed in light-protected conditions and at 4 °C in order to limit loss of biological activity.

### 3.4. Fractionation of the Hydroalcoholic Extract

Lyophilized and finely powered tissue samples of *Cassiopea andromeda* (umbrella and oral arms) were subjected to the fraction’s separation protocol, as described in Leone et al. (2013) [4], by a cold-induced acetonitrile/water (ACN/H_2_O) phase separation (Figure 5).

Briefly, each dried extract was weighed, and 1 mL of iced acetonitrile/H_2_O 1:1 (*v*/*v*), was added to 50 mg of dried extract. After stirring by using Vortex mixer, the suspension was left on ice in order to facilitate a phase separation. The ACN-H_2_O phase separation drives complex mixtures of compounds to separate into three distinct phases [91]. After 30 min on ice, the equilibrium resulted in a lipophilic upper phase (UP), a semi-solid intermediate phase, and a hydrophilic lower phase (LP), as shown in Figure 6. The separated fractions were then better separated by centrifugation at 9000× *g* (15 min at 4 °C). All fractions were stored at −20 °C until analysis.

### 3.5. Spectroscopic Analysis

Samples were analyzed by an ultraviolet–visible (UV–vis) spectrophotometers to measure absorbance and fluorescence under UV and visible light. Analyses of multiple samples were performed in parallel using a 96-well microplate (Corning) and the Infinite M200, quad4 monochromator™ detection system (Tecan, Männedorf, Switzerland). The spectrophotometer detection parameters were set to detect UV–vis or fluorescence absorbencies following the manufacturer’s instructions. Blanks with ethanol, acetonitrile, acetonitrile:H_2_O, or H_2_O were used.

### 3.6. Protein Content

The protein concentration was estimated in UPs and LPs by Bradford assay [92] using bovine serum albumin (BSA) as standard. The assay was modified and adapted to a 96-well microplate (Corning) and analyzed by the Infinite M200, quad4 monochromator™ detection system (Tecan, Männedorf, Switzerland).

### 3.7. Phenol Content

The content of total phenols in UPs and LPs was determined by a modified Folin–Ciocalteu colorimetric method [93]. The test solutions containing 50 μL of sample were mixed with 50 μL (1:4) of Folin–Ciocalteu phenol reagent and with 100 μL of 0.35 M sodium hydroxide (NaOH). After 5 min, at room temperature in the dark, the absorbance was spectrophotometrically measured at 720 nm. The calibration curve was plotted versus concentrations of gallic acid ranging from 0 to 40 μg/mL, which were used as standard. The results were expressed as gallic acid equivalents (GAE) per gram of dried hydroalcoholic extract.

### 3.8. In Vitro Antioxidant Capacity Assay

The total antioxidant activity in UPs and LPs was assayed by TEAC (Trolox equivalent antioxidant capacity) method [94] based on the scavenging of the blue/green ABTS radical [2,20-azinobis-(3-ethyl-benzotiazolie-6-sulfonic acid)], which was converted into a colorless product. The assay was adapted to a 96-well microplate (Corning) for Infinite M200. Appropriate blanks with the relative solvent were run in each assay, and a Trolox calibration curve was prepared under the same conditions of the samples. Briefly, 10 µL of each sample was added to 200 µL of ABTS+ solution and stirred, and the absorbance at 734 nm was read at 6 min. The antioxidant activity value was expressed as nmol of Trolox equivalent (TE) per g of jellyfish tissue and as nmol TE per g of hydroalcoholic extract (nmol TE/g of ExDW).

### 3.9. Symbiont Quantification

A known amount (ranging from 20 to 50 mg) of the lyophilized powder of the oral arms and umbrellas of the *C. andromeda* jellyfish was resuspended in four volumes of filtered seawater, vortexed, and stirred with a micromagnet for 20 min. The total volume of the suspension was recorded, and the determination of the symbiont concentration was carried out within 1 h using a Countess 3 Automated Cell Counters (Thermo Fisher Scientific Inc., Waltham, Massachusetts, USA) following the manufacturer’s instructions. Autofluorescent cells were detected, automatically counted, and compared to the dry weight of the lyophilized tissues. Representative images of the samples were also captured by laser scanning confocal microscope (Zeiss LSM 5 Pascal) and ZEN software (Zeiss). Images were detected by a Plan-Neofluar 40 × 0.75 objective, excitation laser at 488 nm (9.0%), emission windows 505–530 nm band pass (BP) filter, and 660 nm long pass (LP) filter.

### 3.10. Lipid Extraction and Fatty Acid Identification in UP

The lipophile fraction (upper phase, UP) was analyzed for lipid content and fatty acid identification. The UP coming from the fractionation protocol (Section 3.4) was evaporated under a gaseous nitrogen flow and immediately esterified and analyzed for lipid composition by GC-MS analysis, as already reported [7].

### 3.11. Pigment Identification and Quantification

The determination of the pigment content was carried out by HPLC analysis for each sample, both on the hydroalcoholic extract (ExDW) and on the fractions obtained, namely the lipophilic (UP) and the hydrophilic (LP) fractions.

The pigments were separated, according to the protocol used by Guaratini et al. (2009) [95], using an Agilent 1100 HPLC instrument, and analyses were conducted as described by Fraser et al. (2000) [96], with some modifications [4]. Pigment separation (Appendix A) was carried out using a reverse-phase C30 column (YMC Carotenoid, 5 mm, 250 × 4.6 mm) (YMC Inc., Wilmington, NC, USA) maintained at constant pressure and at the constant temperature of 25 °C. The mobile phase, consisting of methanol (A), 0.2% ammonium acetate in 80% methanol solution (20/80 *v*/*v*) (B), and methyl-butyl-ether (C), was used with a flow rate of 1 mL/min and with the following isocratic gradients: from 0 to 12 min 95% A and 5% B; from 12 to 42 min 80% A, 5% B, and 15% C; from 42 to 62 min, 30% A, 5% B, and 65% C; from 62 to 70 min 95% A and 5% B. The injection volume was 10 µL. The column was rebalanced for 10 min between runs. The absorbances were recorded by a diode array (Agilent) at different wavelengths specific for each selected pigment (λ = 450 nm for diadinoxanthin, peridinin, and lutein; λ= 475 nm for b-carotene; and l = 654 nm for chlorophyll-a). For the identification of each pigment, the UV–visible spectrum and the retention time on the chromatogram were analyzed in each sample, which were then compared with properly purchased standards (ChromeDex, Los Angeles, CA, USA) dissolved in ethyl acetate. The quantifications were carried out by comparing the areas of each pigment with the relative calibration line obtained by injecting known and increasing concentrations of each standard.

### 3.12. Statistical Analysis

Data were analyzed by statistical program Graph Pad Prism v5.0. Umbrella and oral arms extracts and phases were compared in yields, protein, phenols, and antioxidant activity using a two-tail unpaired *t*-test (alpha= 0.05); meanwhile, a two-way ANOVA followed by Bonferroni or Tukey’s multiple comparison post hoc tests was used to compare the two groups by pigments composition and phenols content, respectively.

## 4. Conclusions

Our previous studies on Mediterranean outbreak-forming jellyfish suggested that jellyfish biomasses, usually considered a nuisance for human activities at sea, could represent a valuable source of functional food or value-added compounds for nutraceutical, cosmeceutical, or pharmaceutical uses. The increasing presence in the Mediterranean Sea of non-indigenous species such as *Cassiopea andromeda* opens new perspectives in jellyfish biomass exploitation. The zooxanthellate jellyfish *Cassiopea andromeda* is a mixotrophic and easily farmable holobiont that could provide exploitable biomass in a sustainable way. The valuable biochemical composition of *C. andromeda* stimulated further development of extraction protocols and compositional analyses, showing that a significant amount of beneficial fatty acids, phenolic compounds, and pigments can be readily extracted from the biomass of the holobiont *C. andromeda*/dinoflagellate, while the remaining biomass still contains valuable proteinaceous compounds. Studies of biological activities linked to these fractions in cell cultures are now in progress.

In view of its possible future large-scale breeding, the upside-down jellyfish deserves attention, as it may represent at the same time a sustainable potential new exploitable resource of bioactive compounds and a potential food/feed resource not yet fully investigated. The recognized value of bioactive compounds with established antioxidant properties and beneficial effects on human health, along with the emergence of eco-friendly extraction techniques, positions this system as a bio-factory for pioneering approaches in the green-blue bioeconomy.

## Figures and Tables

**Figure 1 marinedrugs-21-00272-f001:**
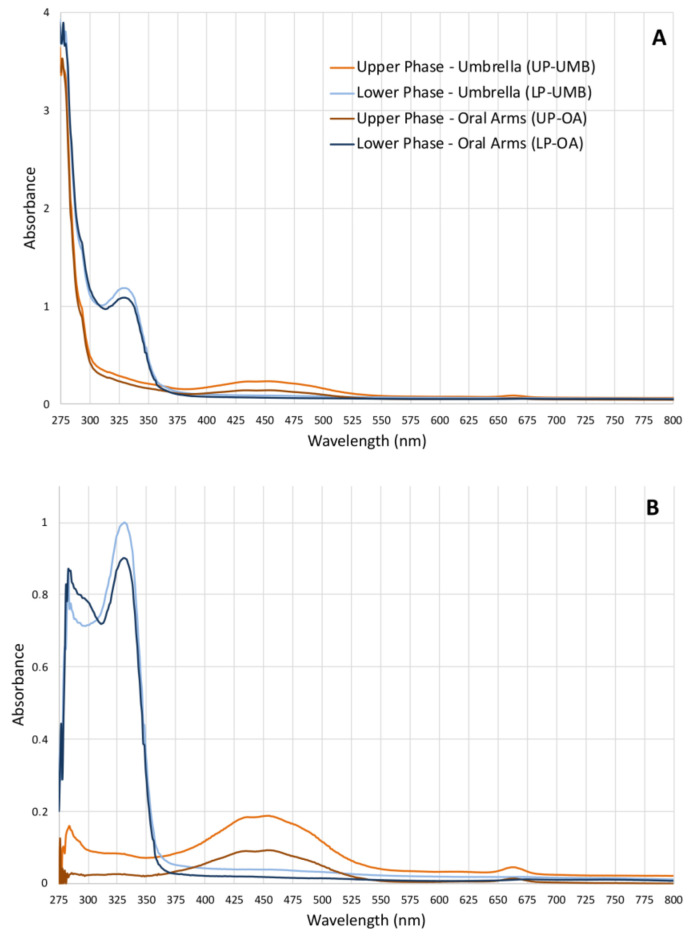
(**A**) Graphs of the absorbance spectra of the upper phases (UPs) and lower phases (LPs) fractionated from the *C. andromeda* extracts umbrella (UMB) and oral arms (OA) measured from 230 to 1000 nm (wavelength step size 1 nm) in polar organic solutions. (**B**) Absorbance of the extract-subtracted fractions of the absorbance of acetonitrile for UPs and water/acetonitrile for LPs used as solvents.

**Figure 2 marinedrugs-21-00272-f002:**
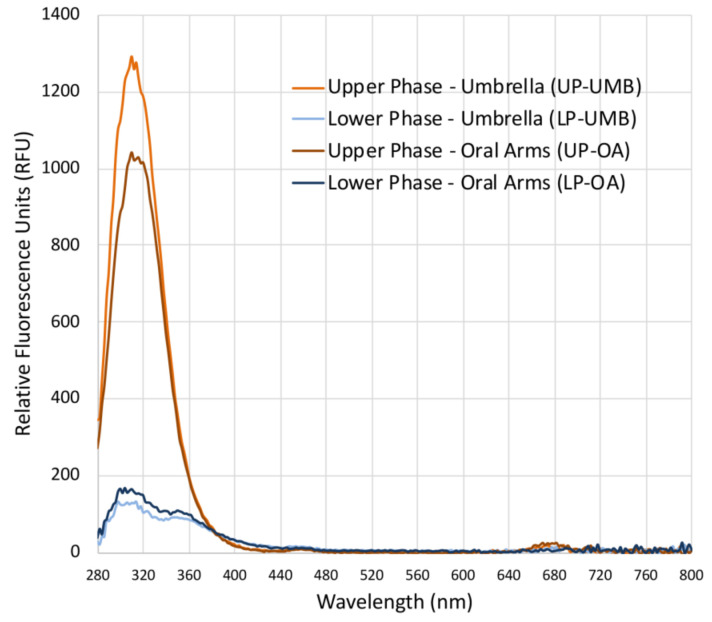
Graph of the fluorescence spectra of the upper phases (UPs) and lower phases (LPs) of the *C. andromeda* extracts from the umbrella (UMB) and oral arms (OA), measured in the range from 280 to 850 nm (excitation wavelength 230 nm, wavelength step size 2 nm) in polar organic solutions. Blanks were acetonitrile for UP and water/acetonitrile for LPs.

**Figure 3 marinedrugs-21-00272-f003:**
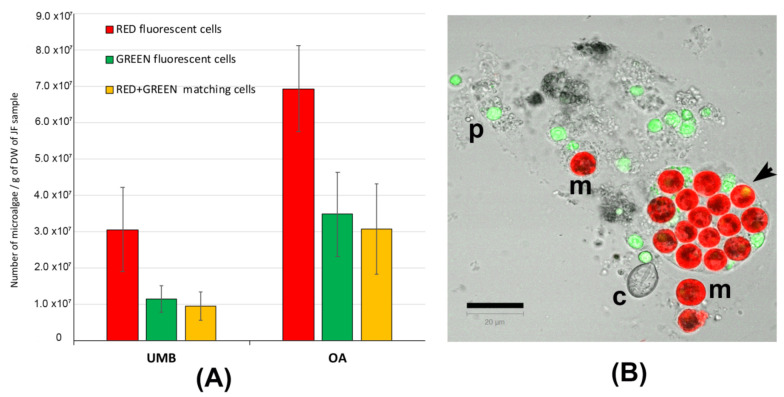
(**A**) Graphs of the number of microalgae (zooxanthellae) measured in the tissues of umbrellas (UMB) and oral arms (OA) of *C. andromeda*. (**B**) Confocal microscope image of the lyophilized jellyfish tissue resuspended in seawater. Red-autofluorescent chloroplasts in microalgae (m), green-autofluorescent particles (p), and a cnidocyst (c) are visible. The autofluorescent particles are automatically counted as red, green, and red + green autofluorescence. The round-shaped red-autofluorescent cells with a diameter around 10 μm were considered as microalgal cells. Bar = 20 μm.

**Figure 4 marinedrugs-21-00272-f004:**
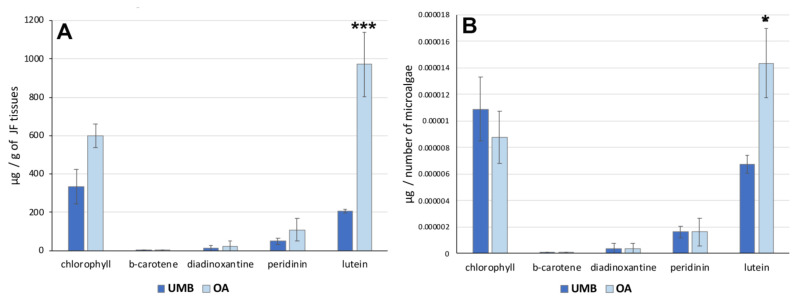
(**A**) Graphs showing the concentration of pigments in the tissues of the umbrella (UMB) and oral arms (OA) of *C. andromeda* as expressed as mg per mg of lyophilized tissue (**A**) and per number of microalgae (**B**). Statistical analysis: two-way ANOVA test (*p* < 0.05) followed by Bonferroni test. Significant differences between UMB or OA are indicated with * for values of *p* < 0.05 and with *** for *p* < 0.001.

**Figure 5 marinedrugs-21-00272-f005:**
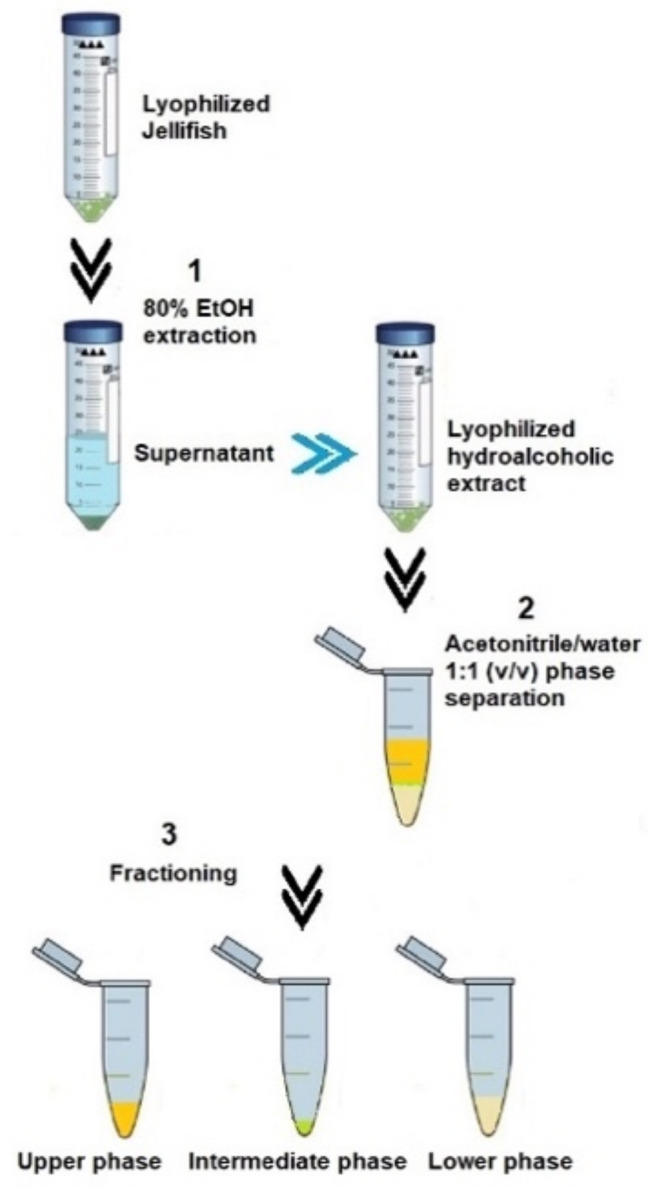
Steps of extraction and phase separation from the lyophilized jellyfish samples following the Leone et al. (2013) [4] methodology.

**Figure 6 marinedrugs-21-00272-f006:**
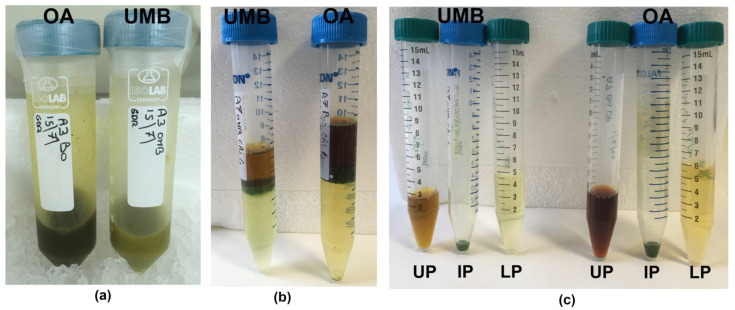
Steps of extraction and fractionation of the hydroalcoholic extracts (**a**) from the umbrella (UMB) and oral arms (OA) of *C. andromeda* jellyfish. Phase separation after ACN/H_2_O precipitation (**b**) and separated phases (**c**). UP, upper phase; IP, intermediate phase; LP, lower phase.

**Table 1 marinedrugs-21-00272-t001:** Dry weight and yield of the hydroalcoholic extracts from the umbrella and oral arms of different specimens of *C. andromeda*. DW, dry weight; ExDW, dried hydroalcoholic extract; SD, standard deviation. The data were statistically analyzed with a *t*-test, two-tail alpha = 0.05.

	UMBRELLA (UMB)			ORAL ARMS (OA)		
Specimen	DW (g)	ExDW (g)	Yield (%DW)	DW (g)	ExDW (g)	Yield (%DW)
Ca1	0.931	0.382	41.1	1.142	0.560	49.0
Ca2	0.615	0.204	33.2	0.761	0.356	46.8
Ca3	0.636	0.264	41.6	1.059	0.464	43.8
Ca4	0.393	0.164	41.7	0.946	0.408	43.2
Ca5	0.822	0.360	43.8	1.052	0.459	43.6
Ca6	0.852	0.356	41.8	0.950	0.423	44.5
Ca7	0.720	0.336	46.7	1.410	0.630	44.7
Ca8	0.710	0.330	46.5	1.640	0.690	42.1
**Mean ± SD**	**0.710** **± 0.168**	**0.300** **± 0.080**	**42.1** **± 4.2**	**1.120** **± 0.281**	**0.499** **± 0.116**	**44.7 ± 2.2**

**Table 2 marinedrugs-21-00272-t002:** Protein content in the upper phase (UP) and lower phase (LP) from the umbrella (UMB) and oral arms (OA) of eight samples of *C. andromeda*. Data are expressed as mg per g of hydroalcoholic extract dry weight (ExDW) for each sample (*n* = 3) and mean ± standard deviation (SD) was calculated taking all data into consideration. Superscript lowercase letters indicate significant differences; when data are analyzed by unpaired *t*-test, two-tail alpha = 0.05.

	Total Protein Content			
Specimen	UMBRELLA		ORAL ARMS	
	UMB-UP	UMB-LP	OA-UP	OA-LP
	mg of Proteins/ g of ExDW (mean ± SD)			
Ca1	1.94 ± 0.72	4.89 ± 0.78	3.52 ± 0.62	5.52 ± 0.47
Ca2	1.24 ± 0.03	1.53 ± 0.17	2.94 ± 0.18	5.97 ± 0.32
Ca3	2.10 ± 0.29	4.43 ± 0.27	6.98 ± 0.32	6.41 ± 0.64
Ca4	2.36 ± 0.01	3.78 ± 1.10	3.45 ± 0.07	5.26 ± 0.68
Ca5	2.24 ± 0.05	8.35 ± 0.10	6.64 ± 0.49	10.56 ± 0.11
Ca6	2.99 ± 0.06	10.11 ± 0.61	4.49 ± 0.01	13.51 ± 0.34
Ca7	0.34 ± 0.10	2.77 ± 1.41	3.05 ± 1.16	3.02 ± 0.31
Ca8	0.45 ± 0.35	3.74 ± 0.56	3.59 ± 0.09	3.41 ± 0.32
**Mean ±SD**	**1.71 ± 0.94 ^a^**	**4.95 ± 2.87 ^b^**	**4.33 ± 1.60 ^a^**	**6.71 ± 3.58 ^b^**

**Table 3 marinedrugs-21-00272-t003:** Content of phenolic compounds in each jellyfish sample assayed in the upper phases (UP) and in the lower phases (LP) from both the umbrella (UMB) and oral arms (OA) of *C. andromeda* jellyfish. Data are expressed as µg of gallic acid equivalents (GAE) per g of hydroalcoholic extract dry weight and are mean ± standard deviation (*n* = 3). All data in each specimen are analyzed by two-way ANOVA test with *p* < 0.05 and Tukey’s multiple comparison post hoc test; superscript lowercase letters indicate significant differences between samples; *** indicate significant differences between UP and LP in UMB or OA tissues.

	Total Phenol Content			
Specimen	UMBRELLA		ORAL ARMS	
	UMB-UP	UMB-LP	OA-UP	OA-LP
	mg GAE/ g of ExDW (mean ± SD)			
Ca3	615.3 ± 3.2 ^a^	1993.7 ± 67.4 ^c^ ***	970.3 ± 25.1 ^b^	4169.9 ± 33.1 ^d^ ***
Ca4	889.8 ± 20.9 ^a^	1830.9 ± 6.5 ^c^ ***	1132.7 ± 9.4 ^b^	3164 ± 27.9 ^d^ ***
Ca5	296.7 ± 6.7 ^a^	1415.5 ± 142.3 ^c^ ***	758.9 ± 13.9 ^b^	2886.1 ± 108.3 ^d^ ***
Ca6	362.3 ± 4.0 ^a^	1350.9 ± 45.2 ^c^ ***	571.9 ± 4.2 ^b^	2573.7 ± 11.9 ^d^ ***
Ca7	919.7 ± 234.6 ^a^	10,518.5 ± 120.1 ^c^ ***	2208.9 ± 25.1 ^b^	11,429.4 ± 266.8 ^d^ ***
Ca8	1248.3 ± 104.5 ^a^	4043.8 ± 118.1 ^c^ ***	2712.8 ± 14.5 ^b^	15,234.0 ± 258.8 ^d^ ***

**Table 4 marinedrugs-21-00272-t004:** Antioxidant activity measured in the upper (UP) and in the lower (LP) fractions from hydroalcoholic extract of the umbrella (UMB) and oral arms (OA) of *C. andromeda*. Data are expressed as nmol of Trolox equivalents (TE) per g of dry weight (DW) of lyophilized jellyfish tissue of UMB or OA and as TE per hydroalcoholic extract dry weight (ExDW); values are mean ±standard deviation (*n* = 6). Superscript lowercase letters indicate significant differences; when data are analyzed by unpaired *t*-test, two-tail alpha = 0.05.

	Antioxidant Activity (AA)			
AA	UMBRELLA		ORAL ARMS	
	UMB-UP	UMB-LP	OA-UP	OA-LP
	Mean ± SD			
nmol TE/g of jellyfish tissue	639.2 ± 471.6 ^a^	5989.2 ± 4341.1 ^b^	1191.4 ± 720.8 ^a^	10,736.3 ± 7728.9 ^b^
nmol TE/g of ExDW)	1514.5 ± 1143.3 ^a^	14126.7 ± 10288.7 ^b^	2583.0 ± 1555.8 ^a^	22,925.2 ± 16,029.5 ^b^

**Table 5 marinedrugs-21-00272-t005:** Composition of fatty acids in the lipophilic fraction (UP) obtained by fractionation of hydroalcoholic extracts from *Cassiopea andromeda* jellyfish. Data are mean of three independent experiments and are expressed as percentage of the total fatty acids.

Fatty Acids in the Lipophilic Fraction of 80% Ethanol Extract of *Cassiopea andromeda* Jellyfish	
Fatty Acid (FA)	Upper Phase (UP) from 80% Ethanol Extract Separation
*Saturated FA (SFA) %*	
Lauric acid *C12:0*	8.5 ± 0.8
Myristic acid *C14:0*	2.5 ± 0.3
Palmitic acid *C16:0*	25.2 ± 2.5
Stearic acid *C18:0*	3.6 ± 0.4
Arachidic acid *C20:0*	-
**Total SFA (%)**	**39.8 ± 4.0**
*Mono-unsaturated FA (MUFA) %*	
Palmitoleic acid *C16:1* (ω7)	2.8 ± 0.3
Oleic acid *C18:1 cis*-9 (ω9)	-
Isoleic acid *C18:1 trans-10*	3.0 ± 0.3
**Total MUFA (%)**	**5.8 ± 0.8**
*Polyunsaturated FA (PUFA) %*	
Linoleic acid *C18:2 cis-9,12* (ω6)	-
Isolinoleic acid *C18:2 cis-6,9* (ω9)	-
Linolenic acid *C18:3 cis-9,12,15* (ω3)	2.4 ± 0.3
Stearidonic acid *C18:4* (ω3)	16.3 ± 1.6
Eicosadienoic acid *C20:2* (ω6)	2.7 ± 0.3
Arachidonic acid *C20:4* (ω6)	14.4 ± 1.5
Eicosapentaenoic acid *C20:5* (ω3)	2.4 ± 0.3
Docosatetraenoic acid *C22:4* (ω6)	-
Docosapentaenoic acid *C22:5* (ω3)	2.2 ± 0.2
Docosahexaenoic acid *C22:6* (ω3)	14.0 ± 1.4
**Total PUFA (%)**	**54.4 ± 5.5**
**Total fatty acids**	**100**
*Σω6*	*17.2*
*Σω3*	*37.3*
**Ratio *ω*6/*ω*3**	** *0.5* **

**Table 6 marinedrugs-21-00272-t006:** Concentration of some zooxanthellae pigments in the total extract DW (ExDW) as expressed in mg/g of ExDW and in the upper phases obtained from the umbrella (UMB-UP) and oral arms (OA-UP) of *C. andromeda* jellyfish expressed as mg/g of UP-DW. Superscript lowercase letters indicate significant differences between UMB and OA samples when data are analyzed by two-way ANOVA test with *p <* 0.05 and Bonferroni post hoc test.

	Hydroalcoholic Extract (ExDW)		Lipophilic Fraction of ExDW	
Pigment	UMB	OA	UMB-UP	OA-UP
	μg/g of ExDW		μg /g of UP-DW	
Chlorophyll-a	722.3 ± 205.0 ^a^	1960.6 ± 601.3 ^b^	4420.3 ± 3178.9	2157.0 ± 574.5
β-carotene	8.0 ± 0.7	7.6 ± 1.7	n.d.	28.4 ± 0.2
Diadinoxanthin	26.5 ± 27.5	63.5 ± 28.6	106.4 ± 149.7	285.0 ± 68.4
Peridinin	107.9 ± 37.9	324.5 ± 42.6	838.6 ± 1179.7	1683.6 ± 275.1
Lutein	441.5 ± 15.1 ^c^	3066.4± 425.0 ^d^	2809.1 ± 3955.1	8978.0 ± 876.7

## Data Availability

Relevant data are contained within the article; raw data are available from the corresponding author, A.L.

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
