# Peer review of "The Zooxanthellate Jellyfish Holobiont Cassiopea andromeda, a Source of Soluble Bioactive Compounds"

_marinedrugs, 2023, doi:10.3390/md21050272_

Round 1
Reviewer 1 Report
Comments and Suggestions for Authors
In present manuscript, the authors aim to evaluate the biochemical profile of the hydroalcoholic extract of the zooxanthellate jellyfish, but the extraction was conducted on the jellyfish tissues that containing the zooxanthellate. Instead, the authors shall isolate and cultivate the zooxanthellate prior to chemical extraction.
L183: What is the purpose of conducting spectroscopic analysis when the results are not conclusive?
L222: The authors evaluate protein content of the extract instead of the dried biological samples, indicates that the authors did not consider insoluble proteins. Thus, present result on protein analysis is not representable for both UMB and OA, yet misleading. Instead, protein analysis should be conducted directly on the dried biological samples.
L244: Phenolic compounds usually are semi-polar and non-polar, but the extracts used for present study is from 80% ethanol, which is relatively polar and less suitable for phenolic compounds. Why the authors decided to determine phenolic compounds from polar extract instead of others?
L286: Missing of positive control in antioxidation assay.
L324: Proper approach for fatty acid analysis is on chloroform extracts, not the extracts originated from 80% ethanol. Again, the authors ignored the present of semi-polar and non-polar lipids, which will have richer fatty acid profile after derivatization.
L423: Present results are not conclusive as the authors only focus on 80% ethanolic extracts.
L550: Please amend the subtitle since the Total Lipid Extraction is not conducted.
L559: Please include the labeled chromatograms in supplementary.
Author Response
Dear Reviewer,
thank you for your revision, here are the answers to your comments hoping to better explains and clarify the aim of our work.
Reviewer. In present manuscript, the authors aim to evaluate the biochemical profile of the hydroalcoholic extract of the zooxanthellate jellyfish, but the extraction was conducted on the jellyfish tissues that containing the zooxanthellate. Instead, the authors shall isolate and cultivate the zooxanthellate prior to chemical extraction.
Response. Concerning your comment on the suggestion to isolate zooxanthellae before chemical extraction, we would like to clarify that this is not in line with the purpose of this paper and we thank you as we may have the opportunity to explain that.
In our work we want to characterize the entire biomass of C. andromeda intended as “holobiont” in order to evaluate the possible use of this zooxanthellate species for the extraction of several valuable compounds. Indeed, since C. andromeda holobiont includes the animal tissues of jellyfish and its associated microorganisms consisting of bacteria, archaea, fungi, viruses and protists including the dinoflagellate alga Symbiodinium, which contribute to the synthesis of different molecules. Since we were interested in evaluating the composition of this complex biomass, we started from our previous work (De Rinaldis et al., 2021) and here we continue by characterizing the fractions obtained from the hydroalcoholic extract.
To clarify this point, we modified the title and the abstract and tried to highlight it in the last paragraph of the Introduction section, in the first paragraph of the result section and throughout the text (see text in red).
Reviewer. L183: What is the purpose of conducting spectroscopic analysis when the results are not conclusive?
Response. This is a study on extraction protocols and compound characterization in newly obtained fractions (UP and LP) from hydroalcoholic extract of C. andromeda, then a first simple qualitative spectrophotometric and fluorometric analyses were carried out in order to detect different class of compounds in a cheaper, faster, and thus more accessible methods than more specific analyses such as analytical chromatography techniques (e.g., high-performance liquid chromatography). In addition, the comparison with the known characteristic spectra of absorption and fluorescence of known compounds (e.g., chlorophylls, carotenoids, phenols, etc) helps to identify the presence of other compounds and/or contaminants.
Lines 189-192: The following sentence was added to the manuscript in order to clarify this point.
“Given the lack of literature data on the possible compounds present in the hydroalcoholic extract fractions, and of related analyses for their identification, a simple qualitative analysis of the upper phase (UP) and the lower phase (LP) was first per-formed to rapidly identify compound categories rather than individual compounds.”
Reviewer. L222: The authors evaluate protein content of the extract instead of the dried biological samples, indicates that the authors did not consider insoluble proteins. Thus, present result on protein analysis is not representable for both UMB and OA, yet misleading. Instead, protein analysis should be conducted directly on the dried biological samples.
Response. Thank you for highlighting this point which was perhaps unclear. The evaluation of the protein content in the dried biological sample had already been published in De Rinaldis et al. (2021), where protein quantification was performed on the whole biomass including hydroalcoholic soluble and insoluble proteins. In the present work we intend to deepen the study of the hydroalcoholic extract by fractionation and characterization of the lipophilic and hydrophilic fractions (UP and LP).
Please see lines 131-135 and lines 235-237.
Reviewer. L244: Phenolic compounds usually are semi-polar and non- polar, but the extracts used for present study is from 80% ethanol, which is relatively polar and less suitable for phenolic compounds. Why the authors decided to determine phenolic compounds from polar extract instead of others?
Response. Phenol compounds, which include polyphenols, comprise a huge family of compounds with diverse structures. Solubility depends on the polar properties of the polyphenols and also on the presence or absence of the glycoside moiety since many phenolic compounds are present in the glycoside form. At the best of our knowledge, polyphenols are generally more hydrophilic than lipophilic due to their phenolic nature. Therefore, free polyphenols along with aglycones, glycosides and oligomers can be easily extracted by solvents such as methanol, ethanol, acetonitrile and acetone, or by their mixtures with water.
In our study we used water and ethanol as solvents, one of the more conventional methods for extracting phenolic compounds, instead of environmental hazardous solvents such as methanol, acetone, acetonitrile, ethyl acetate, dichloromethane, hexane, petroleum ether etc, since one of our purposes is also to develop green-solvent based protocols.
Based on this, in previous works we got good yields in phenolic compound by extraction with 80% ethanol (see Leone et al., 2015; De Rinaldis et al., 2021). In this work, we continued the characterization of this extract by fractionation and analytical characterization of the fractions.
We tried to clarify this along the text.
Reviewer. L286: Missing of positive control in antioxidation assay.
Response. The reviewer is correct in considering a positive control for the antioxidant activity assay. However, in TEAC analysis, Trolox, which is a standard synthetic antioxidant, is usually used as a positive control and the solvents alone and a no-sample extraction as blank and negative controls, respectively.
Reviewer. L324: Proper approach for fatty acid analysis is on chloroform extracts, not the extracts originated from 80% ethanol. Again, the authors ignored the present of semi-polar and non-polar lipids, which will have richer fatty acid profile after derivatization.
Response. Thanks to the reviewer for offering the opportunity to clarify this point and the used procedure. A first fatty acids characterization, based on methanol:chloroform method (Bling and Dyer, 1959) was already carried out on both full samples of dried UMB and OA and on their 80% ethanol extract (methanol:choloform method applied on dried 80% ethanol extract) and the results were already published in our previous work De Rinaldis et al., 2021 (https://doi.org/10.3390/md19090498).
As the whole biomass was already analysed in the previous work, in this work we focused on the characterization of the hydroalcoholic extract (80% ethanol) by its fractionation and fraction’s analysis. The 80% ethanol is able to extract a complex mixture of compounds including some semi-polar and non-polar lipids such as free carotenoids; since the fractionation method with acetonitrile/water is able to segregate the lipophilic compounds in the UP fraction, this last was directly analysed by GC-MS for FA composition analysis and by HPLC for pigment analysis.
For information of the reviewer, we focused on the characterization of hydroalcoholic extract and on their fractions as they demonstrated interesting biological activity on human cell culture systems in parallel experiments. The results related to biological activity are part of another work and the characterization here carried out is important for the interpretation of that results. Since the amount of results was difficult to place in a single paper, we decided to divide the work into a part of biochemical characterization and one of analysis of biological activity (submission stage).
We have summarized this point in lines 349-351 and lines 355-357.
Reviewer. L423: Present results are not conclusive as the authors only focus on 80% ethanolic extracts.
Response. Considering the characterization of C. andromeda biomass already published in our previous work, the results presented here have deepened the biochemical description of this holobiont leading to a better description of the whole biomass.
Since one of the objectives of our work is to develop protocols based on green solvents for the extraction/isolation of natural bioactive compounds, in our opinion the choice of 80° ethanol extraction represents a suitable first approach. On the basis of environmental and health and safety issues, ethanol and acetone are preferred solvents, compared to hexane, diethyl ether, dichloromethane and chloroform, which are generally used for extraction of carotenoids. Moreover, the use of ethanol 80% is a good compromise in order to extract both chlorophylls and carotenoids, as widely reported in literature (Saini & Keum, 2018; Hosikan et al., 2010).
In addition, the extraction with ethanol solutions allowed to maintain the antioxidant activity, assessed by the TEAC. Moreover, the extraction yield is generally higher with the polar solvent ethanol, when compared to hexane.
Finally, in this work we intended to characterize the fractions obtained from the ethanolic extracts, because they were active on cell culture systems (paper in submission stage).
We have tried to clarify throughout the text and in lines 115 – 129 and 131-135.
References
Saini, R. K., & Keum, Y. (2018). Carotenoid extraction methods: A review of recent developments. Food Chemistry, 240, 90-103. https://doi.org/10.1016/j.foodchem.2017.07.099)
Aris Hosikian, Su Lim, Ronald Halim, Michael K. Danquah, "Chlorophyll Extraction from Microalgae: A Review on the Process Engineering Aspects", International Journal of Chemical Engineering, vol. 2010, Article ID 391632, 11 pages, 2010. https://doi.org/10.1155/2010/391632.
Reviewer. L550: Please amend the subtitle since the Total Lipid Extraction is not conducted.
Response. Done.
Reviewer. L559: Please include the labeled chromatograms in supplementary.
Response. Done. Supplementary Figure S3 has been added.
Reviewer. Quality of English Language (x) Extensive editing of English language and style required.
Response. The text was revised for English language and style.

Reviewer 2 Report
Comments and Suggestions for Authors
The abstract objectively represented the article; it did not contain results not presented and substantiated in the main text and did not exaggerate the main conclusions.
Introduction. The introduction places the study in a broad context and highlights its importance. It defined the purpose of the work and its significance. However, the specific hypotheses being tested were not included. The current state of the research field was reviewed carefully, and critical publications were cited.
Suggestions:
- Highlight controversial and diverging hypotheses
- Highlight the main conclusions
- It could be shorter, using the most relevant references. There are more than one hundred references. Please check.
The results provided a concise and precise description of the experimental results and interpreted them from the perspective of previous studies.
Suggestions
- It should be interesting to add future research directions.
- Please check 2.5 in the result section. It needed to be more obvious why it was discussed after antioxidant activity.
The materials and methods were described sufficiently to allow others to replicate and build on published results. The hydroalcoholic extraction and fractionation of this extract protocol were described in detail. Moreover, well-established methods were briefly described and appropriately cited.
Suggestion
- Please review section 3.8; it needed to be clarified why Symbiotic quantification was included in this section.
The conclusion section should be rewritten; it is too long. Moreover, avoid including references. Please highlight its importance and relevance. Also, it should be interesting to address the implications for further research or action.
Author Response
Dear Reviewer,
thank you for your revision.
Reviewer. The abstract objectively represented the article; it did not contain results not presented and substantiated in the main text and did not exaggerate the main conclusions.
Introduction. The introduction places the study in a broad context and highlights its importance. It defined the purpose of the work and its significance. However, the specific hypotheses being tested were not included. The current state of the research field was reviewed carefully, and critical publications were cited.
Suggestions:
- Highlight controversial and diverging hypotheses.
- Highlight the main conclusions.
- It could be shorter, using the most relevant references. There are more than one hundred references. Please check.
Response. Thanks for the suggestions, in the last paragraph of the introduction the working hypothesis and the purpose of the experimental approach have been clarified, some not relevant sentences have been eliminated in the introduction, the references have been checked and their number has been reduced.
Reviewer. The results provided a concise and precise description of the experimental results and interpreted them from the perspective of previous studies.
Suggestions
- It should be interesting to add future research directions.
- Please check 2.5 in the result section. It needed to be more obvious why it was discussed after antioxidant activity.
Response. Thank you, for suggestions. A paragraph at the end of the section 2.4 was added with the aim to clarify and introduce the section 2.5 and to make explicit the rationale of the order of the experiments. In addition, some possible future applications were briefly mentioned at the end of the Result and Discussion section.
Reviewer. The materials and methods were described sufficiently to allow others to replicate and build on published results. The hydroalcoholic extraction and fractionation of this extract protocol were described in detail. Moreover, well-established methods were briefly described and appropriately cited.
Suggestion
- Please review section 3.8; it needed to be clarified why Symbiotic quantification was included in this section.
Response. Thank you for your remark, it was a typo, the symbiont quantification is in a separated section, the correct numbering of the sections in the manuscript has been restored.
Reviewer. The conclusion section should be rewritten; it is too long.
Moreover, avoid including references.
Please highlight its importance and relevance.
Also, it should be interesting to address the implications for further research or action.
Response. The conclusion section was revised, the references were eliminated and some sentences were rephrased in order to highlight the importance and relevance and future implications. We hope this has improved the Conclusions section.

Reviewer 3 Report
Comments and Suggestions for Authors
It's an interesting article to understand about the valuable compounds of zooxanthellae of jellyfish. However, authors failed to show the biological action of those valuable compounds. Even if it has high protein or FA, all need to be tested and results need to be discussed here with biological importance of fractions.
Authors must provide few figures of the targeted jellyfish and zooxanthellae.
Authors need to do following corrections.
abstract; Forskal typo. , why oral arms and umbrella are chosen to characterize? Zooxanthellae found in only this jellyfish or many other jellyfish possess? How did you identify the jellyfish?
Results: yield is good, but why have taken too little? would have collected more number of jellyfish and increase in weight! Why ethanol is used, not butanol or any other solvent?
L246: what is UP e LP? what is e?
Biological activities of these fractions need to be added, then only this paper has a value, otherwise, it is not a rich data.. Fractions and its biological activities subtitle is better to add and discuss. MS title is source of valuable compounds and phenol and antioxidant compounds are revealed, in addition to that biological action will give more interesting for the readers to understand.
Author Response
Dear Reviewer,
thank you for your revision.
Reviewer. It's an interesting article to understand about the valuable compounds of zooxanthellae of jellyfish. However, authors failed to show the biological action of those valuable compounds. Even if it has high protein or FA, all need to be tested and results need to be discussed here with biological importance of fractions.
Response. The reviewer is right, the biological activity was indeed evaluated on human cell cultures in parallel experiments, however the amount of results was difficult to place in a single paper, then we decided to focus here on the characterization of hydroalcoholic extract fractions as they demonstrated biological activity on human cell cultures. The results related to biological activity are part of another work and the characterization here carried out is important for the interpretation of that results. (Please, see lines 131 - 135)
Reviewer. Authors must provide few figures of the targeted jellyfish and zooxanthellae.
Response. Figures of zooxanthellae from the holobiont C. andromeda were already provided in Supplementary Figures (Figure S1). Figures of C. andromeda specimens in situ and during laboratory handling are provided as additional Supplementary Material (Figure S2).
Reviewer. Authors need to do following corrections. Abstract; Forskal typo,
Response. As far as we know, the most used name of the Swedish naturalist who described C. andromeda is Peter Forsskål, although other spellings are also used (e.g. Forskal).
Reviewer. Why oral arms and umbrella are chosen to characterize?
Response. Thanks to the Reviewer for offering the opportunity to clarify. We are focusing on C. andromeda jellyfish biomass which, due to their ecology and physiology, could represent a sustainable source of bioactive compounds easily extractable with simple protocols based on green solvents. Furthermore, C. andromeda can also be easily farmed.
In our previous work (De Rinaldis et al., 2021) we demonstrated that C. andromeda can be considered a promising source for bioprospecting, its hydroalcoholic extracts, especially from the oral arms, exert a remarkable antioxidant activity. In this work, further sub-fractionations and characterization of the hydroalcoholic extracts - obtained separately from umbrella and oral arms - were carried out to analyse the specific composition in proteins, lipids, pigments, proteins and phenolic compounds of each fraction and each part of the body. The benthonic behaviour and the particular upside-down position of this jellyfish also offers a possible diversity of oral arm and umbrella tissues. (lines 115 -121 and lines 131-135).
Reviewer. Zooxanthellae found in only this jellyfish or many other jellyfish possess.
Response. It’s estimate that 20–25% of Scyphozoa species are zooxanthellate (facultative symbiotic species included) and the suborder Kolpophorae (Scyphozoa: Rhizostomeae) consists, with some few exceptions, of only zooxanthellate jellyfishes. Symbioses between zooxanthellae and medusozoans are often little documented in the literature, we already studied C. tuberculata (Leone et al., 2015) that lives in symbiosis with zooxanthellae.
Reviewer. How did you identify the jellyfish?
Response. The specimens used for this study belongs to the same population that previously sampled and molecularly barcoded (COI marker) by Maggio et al. 2019, and identified as C. andromeda. Appropriate text has been added in Materials and Methods (section 3.2).
Reviewer. Results: yield is good, but why have taken too little? would have collected more number of jellyfish and increase in weight!
Why ethanol is used, not butanol or any other solvent?
Response. After the lyophilization process, C. andromeda biomass (made up of 95-97% of water, like other Scyphomedusae) is about the 3-5% of FW. Before drying we separated each jellyfish in umbrella and oral arms, so the obtained dry mass for each sample was very little but representative of the sample.
Furthermore, we were interested in analyzing each jellyfish individually to verify the variability, we used a suitable number of individuals, so as not to process more animals than necessary also based on the principles of the Nagoya protocol.
Reviewer. L246: what is UP e LP? what is e?
Response. It was a typo, corrected. Thank you.
Reviewer. Biological activities of these fractions need to be added, then only this paper has a value, otherwise, it is not a rich data.
Fractions and its biological activities subtitle is better to add and discuss. MS title is source of valuable compounds and phenol and antioxidant compounds are revealed, in addition to that biological action will give more interesting for the readers to understand.
Response. In this work, only the antioxidant activity tested in vitro was evaluated as first proof of bioactivity. However, other experiments on cancer cells are in parallel to validate the beneficial effects of UP and LP and are matter of another manuscript.
Reviewer. Quality of English Language (x) Moderate English changes required
Response. The text was revised for English language and style.

Round 2
Reviewer 1 Report
Comments and Suggestions for Authors
Overall, the authors have responded all the comments and improved the manuscript accordingly.
Reviewer 3 Report
Comments and Suggestions for Authors
Authors followed all the comments and manuscript is improvised well.